# PROCESS-THEN-RETRIEVE: A MECHANISTIC STUDY OF CROSS-MODAL ALIGNMENT IN VISION-LANGUAGE MODELS

**Arpita Shanbhag, Julia Tran, Dhruv Reddy Mandala, Ayda Sultan**

## ABSTRACT

Despite the capabilities of vision-language models (VLMs), they frequently fail on vision-centric tasks, often tending toward linguistic priors over visual evidence. We hypothesize that this behavior arises from a structural bias toward textual dominance, in which the model prioritizes linguistic evidence over cross-modal integration. Through a mechanistic study of adapter-based VLMs including PALIGEMMA-3B, PALIGEMMA-10B, and QWEN2-VL, we propose a two-phase "Process-then-Retrieve" workflow. While linear probing reveals that semantic visual information is recoverable in early layers, our attention and activation patching experiments demonstrate that the model effectively ignores this data until the final network stages. Instead, early layers are dedicated almost exclusively to textual processing, establishing a strong linguistic trajectory before visual retrieval occurs. We argue that this late-stage integration is a primary contributor to textual dominance, where linguistic priors can override conflicting visual evidence, directly contributing to hallucinations and failures in visual reasoning.

## 1 INTRODUCTION

State-of-the-art vision–language models (VLMs) such as PALIGEMMA integrate large language models with pre-trained vision encoders via projection adapters to achieve strong performance in multi-modal reasoning, but the internal mechanisms that reveal how visual information enters and shapes the linguistic latent space remain a "black box" (Beyer et al., 2024b;a; Shukor & Cord, 2024). Prior work identifies a "modality gap," characterized by geometric separation, late-stage processing, and text dominance, causing models to perform worse on vision-centric reasoning tasks (Shukor & Cord, 2024; Neo et al., 2024; Venhoff et al., 2025b; Nikankin et al., 2025; Wang et al., 2024). Focusing on PALIGEMMA-3B, we show that its "processing-then-retrieval" pipeline causes early layers to prioritize textual context while deferring visual information. We attribute this behavior to the structural constraints of adapter-based architectures, in which the linear projection of visual tokens into the model's reasoning space requires a context-building phase before effective cross-modal integration can occur. While recent work by Venhoff et al. (2025b) identifies late-stage integration as a general phenomenon in VLMs, our work moves beyond observation to identify mechanisms driving this behavior in adapter-based architectures. We validate the existence of the "modality gap" in two ways. First, we mechanistically attribute this delay to a "process-then-retrieve" workflow, in which early layers prioritize text-based context before visual retrieval occurs. Second, through activation patching, we demonstrate that visual processing is largely redundant across subsequent layers for VQA tasks, contradicting the assumption that "visual blindness" is solely an issue of how attention is allocated (Kang et al., 2025).

## 2 EXPERIMENTS

To evaluate whether textual dominance drives failure in vision-centric tasks, we analyze the internal mechanics of PALIGEMMA-3B, PALIGEMMA-10B, and QWEN2-VL. We utilize a Visual Question Answering (VQA) retrieval task where we prompt the model with `"<image> How many objects are in the image?"` using 100-200 valid samples from the PixelProse dataset (Singla et al., 2024). We hypothesize that adapter-based VLMs prioritize linguistic priors over visual evidence by separating procesing streams, leading to a "Process-then-Retrieve" architecture where

visual integration occurs too late to override textual context. **Datasets**: We utilize the PixelProse dataset containing over 16 million image-caption pairs. For each entry, we use the image and its paired caption (VLM-generated or human-made). **Model**: We use PALIGEMMA's SigLIP encoder (448x448 input, 1024 tokens) and a linear adapter that projects image patches into the language decoder's text space. The decoder consists of 18 layers (PALIGEMMA) or 28 layers (QWEN2-VL), each containing transformer blocks that integrate multi-modal information (Beyer et al., 2024b).

## 2.1 VISION REPRESENTATION ACROSS LAYERS

To determine if delayed visual processing enables textual dominance, we track how visual embeddings evolve throughout the network. We compute the cosine similarity between vision-token representations for consecutive layers $l$ and $l + 1$ and averaging it across all vision tokens for 100 samples. As shown in Figure 1, layers 1-10 exhibit high similarity where visual tokens remain static while the model prioritizes textual content. In layers 11-18, there is a sharp drop, which indicates the "retrieval" phase, where visual vectors bridge into the textual space. By the time cross-modal integration happens, the textual stream has already established a strong trajectory that the late injection of visual evidence fails often fails to override. We further analyzed the mechanism driving this

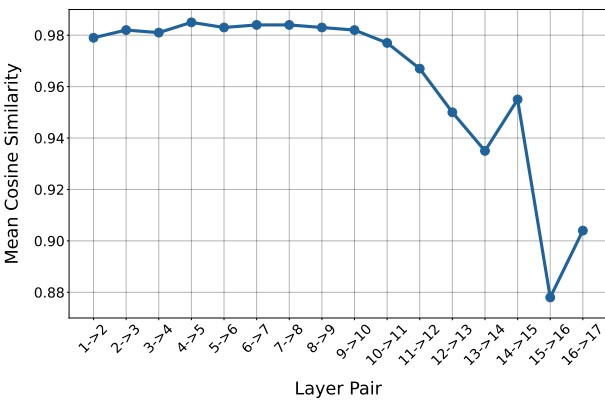

Figure 1: Mean cosine similarity between vision-token representations

late-stage retrieval by decomposing the residual stream. We find that attention contributions play the dominant role in updating visual embeddings (see Appendix B). This confirms that cross-modal alignment is unstable: the model only considers visual evidence by "looking" at the image after the textual reasoning path is largely determined, rather than integrating features from earlier layers.

## 2.2 ATTENTION-OUTPUT ACTIVATION PATCHING

To test if the PALIGEMMA-3B's decision-making is dominated by linguistic priors, we conduct attention-output activation patching experiments. We intervene by intjecting hidden states from self-attention mechanisms of the source layers (early: 2-4, mid: 5-7, late: 10-12) and feed them into the self-attention outputs of the final 3 layers of the model. We evaluated with logit and cross-entropy losses on the target token. Refer to Appendix E for results on QWEN2-VL and PALIGEMMA2-10B.

Vision and textual pathways yield contrasting results. In the vision pathway, patching early states into mid layers yields had minimal effect, indicating that the visual stream is dormant rather than actively refined. However, in the textual pathway, patching early text states into mid and later layers increases the logit change by +6 and significantly reduces loss. This suggests that the answer for the question that the model was prompted with is largely determined by early textual processing. The passivity of the visual stream implies it lacks the causal force to override these strong linguistic priors, directly contributing to failures when visual evidence conflicts with text. Refer to Appendix G for residual stream patching results in PALIGEMMA-3B, QWEN2-VL, and PALIGEMMA2-10B.

Table 1: Attention Output Patching Results

| Category | Patch Direction | Avg Clean Score | Avg Patched Score | Avg Δ Logit | Avg Clean Loss | Avg Patched Loss |
|---|---|---|---|---|---|---|
| Vision | Early → Mid | 12.8346 | 12.9177 | +0.0831 | 14.8269 | 14.8542 |
| Vision | Late → Mid | 12.8374 | 12.8316 | -0.0058 | 14.8260 | 14.8178 |
| Vision | Early → Late | 12.8346 | 12.5501 | -0.2845 | 14.8269 | 14.5287 |
| Vision | Mid → Late | 12.8346 | 12.5695 | -0.2651 | 14.8269 | 14.5158 |
| Text | Early → Mid | 12.8346 | 19.0637 | +6.2291 | 14.8269 | 11.1738 |
| Text | Late → Mid | 12.8346 | 12.3032 | -0.5313 | 14.8269 | 15.6569 |
| Text | Early → Late | 12.8346 | 18.7865 | +5.9519 | 14.8269 | 9.9437 |
| Text | Mid → Late | 12.8363 | 17.3727 | +4.5364 | 14.8272 | 10.1878 |

## 2.3 ATTENTION BY MODALITY PER LAYER

To establish whether textual dominance is caused from a lack of early integration, we analyze how attention is distributed across modalities and layers. For each layer, we extract the attention weights from the final token query and compute the mean attention towards vision and text tokens separately. From this, we track how the PALIGEMMA-3B'S focus on each modality shifts through layers.

Figure 2 reveals the pattern of textual dominance where the model shows a clear reliance on textual tokens in most layers, with visual attention increasing only in the final layers. This indicates that PALIGEMMA defers integrating visual features for the final layers, where cross-modal alignment for answer generation occurs.

Attention decomposition reveals strong modality segregation in query-key interactions. We find that vision queries preferentially attend to vision keys, and respectively, text queries and text keys do the same. Cross-modal attention, from text queries to vision keys required for answering visual questions, is negligible in early layers and becomes significant only in the final transformer blocks. This confirms that visual representations are processed in isolation until late layers, but that they are "retrieved" only when the textual reasoning phase is near completion.

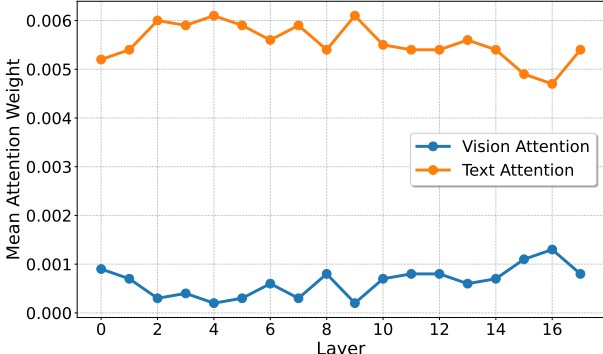

Figure 2: Mean vision vs. text attention weights across layers

## 2.4 LINEAR PROBING GENERALIZATION

We evaluate the robustness and efficiency of multimodal spatial representations across network depth using linear probing. From the PixelProse dataset, we construct a subset of image–caption pairs containing six common object categories (dog, cat, car, person, tree, and building) that frequently participate in spatial relations. We train lightweight linear classifiers on both visual and textual representations by extracting image-token and text-token embeddings from the last four transformer layers, and evaluate them across all layers.

Our analysis investigates how spatial and object-level information is preserved and transformed throughout the network. While we initially hypothesize that visual representations remain relatively stable across layers and that textual representations undergo heavier early-stage processing, our results reveal strong layer-wise specialization. Probe performance is highest when classifiers are evaluated on the same layers on which they are trained, and degrades substantially under cross-layer transfer. This suggests limited representational alignment across depth, highlighting a trade-

off between expressiveness and efficiency that is critical for building adaptive and robust spatial reasoning systems.

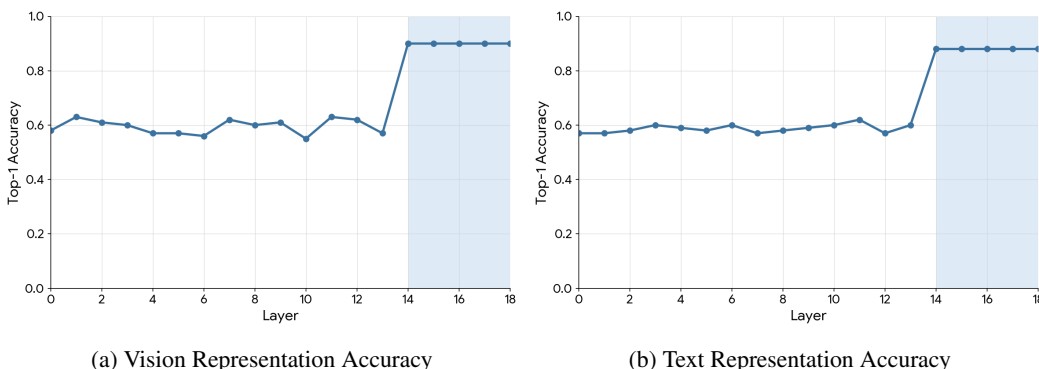

(a) Vision Representation Accuracy    (b) Text Representation Accuracy

Figure 3: Linear probing: Accuracy remains relatively stable across layers (about 55%–63%), followed by a spike in layers 14-18 to around 90% for both vision (left) and text (right) representations.

## 2.5 RESIDUAL STREAM UPDATE

To quantify vision reasoning in adapter-based VLMs, we analyze the total updates to vision-token representations in the residual stream. We compute the difference between pre- and post-layer representations. Figure 4 visualizes the averaged magnitude changes to discover that processing is weakest in early-to-mid layers and spikes in the final layers, suggesting that integration between image and text spaces occurs primarily at output stages.

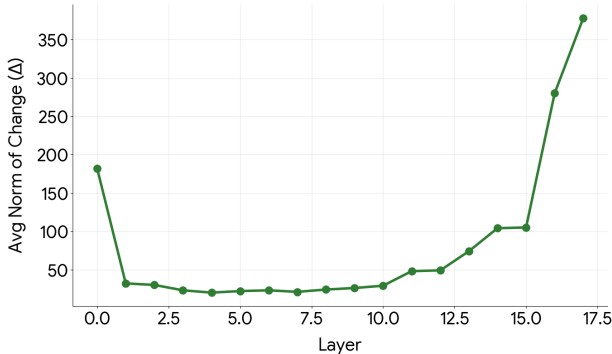

Figure 4: Vision token representation changes: High initial processing at layer 0 before dropping for layers 1-10. Spikes in final layers 15-17.

## 3 CONCLUSION

In this work, we analyzed the mechanistic reasoning behind PALIGEMMA-3B, confirming our hypothesis that adapter-based VLMs typically exhibit early-layer textual data processing behaviors, followed by information retrieval in later layers. Our findings raise fundamental questions about multi-modal representational reasoning in VLMs. The separation of modalities until late layers suggests that current architectures consist of modality-specific processing streams that merge only near output. The dominance of textual attention across the network, with visual attention peaking only in the final layers, suggests an unbalanced focus on language that ultimately limits visual reasoning capabilities. Our linear probing experiments further confirms that models prioritize processing textual features in initial layers, with visual data being used for information retrieval only in later layers, which may not be sufficient time to extract all relevant information into the latent space.

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

## A    OTHER MODELS

For the purposes of output replication, we repeat these experiments on other models in the Appendix. We use QWEN2-VL-2B-INSTRUCT, a pretrained model by Alibaba. Experiments with this model dynamically process images with a patch size of 14x14 pixels, while text input is tokenized systematically through byte pair encoding based on tiktoken (Shantanu, 2025). The model uses a QWEN2-specific Vision Transformer Architecture that is optimized by SwiGLU, which affects activation function (Shazeer, 2020), and RMSNorm, which replaces the traditional layer normalization technique (Jiang et al., 2023b; Bai et al., 2023). This is implemented throughout 28 layers. Additionally, we perform experiments on PALIGEMMA2-10B-PT-448. Similar to PALIGEMMA-3B-PT-448, it uses a SigLIP encoder, but upgrades its text decoder to GEMMA2. (Keysers & Steiner, 2024) This model is used to ensure scalability by increasing PALIGEMMA's parameters.

## B    ATTENTION VS MLP CONTRIBUTIONS

### B.1    PALIGEMMA

In order to understand how much each layer of the multi-modal model, PALIGEMMA, processes vision tokens, we analyze the sum of layer outputs which flows through the transformer, or the residual stream. The attention and MLP sublayers contribute to the vision token representations, as we calculate. This is found by comparing the change in token representations before (pre-contribution) and after (post-contribution) each layer processes the tokens across all hidden states. The scores are extracted from the post-contribution layer. To determine which components drive these changes, we further decompose each layer's residual update by separately measuring the contributions of the attention and MLP sublayers, enabling a direct comparison of their roles in vision-token processing as shown in Figure 5. To determine which of these drives vision processing, we decompose each layer's residual stream contributions by separately passing vision tokens through the attention and MLP sublayers.

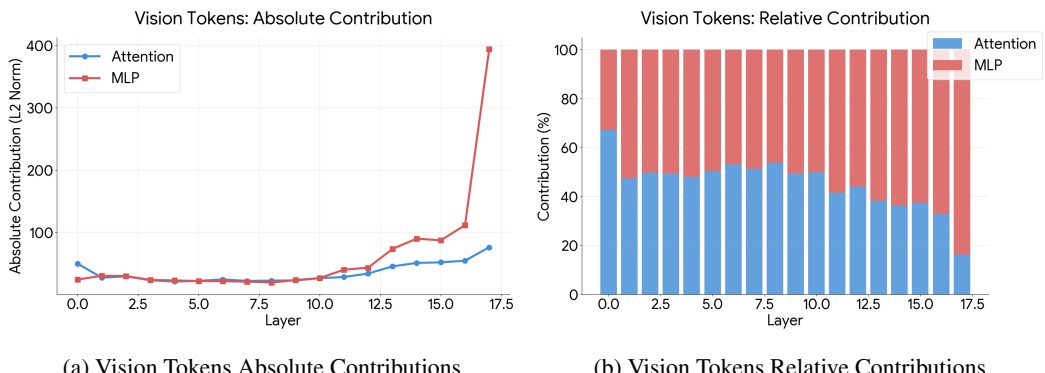

(a) Vision Tokens Absolute Contributions     (b) Vision Tokens Relative Contributions

Figure 5: Attention and MLP contributions to vision token updates: Relative contributions reveal attention and MLP contribute roughly equally throughout most layers, with attention comprising 40-60% of total changes. MLP spikes at the final layer, comprising over 80% of contributions to the residual stream. Absolute contributions demonstrate that the L2 Norm for MLP jumps from around 100 to 400 in the final transformer layer.

Additionally, we analyze the attention and MLP sublayers for text tokens. Figure 6 demonstrates that between these two residual stream contributions, attention consistently increases between layers while MLP spikes dramatically in late layers. However, compared to the absolute total contributions in Figure 5, the contribution is much lower. This alludes to doubts about how attention and MLPs interact with the residual stream in models with more parameters. Further testing should be done

in future experiments to examine how adapter-based VLMs, scaled to large parameter sizes such as 10B and 72B, integrate visual and textual embeddings.

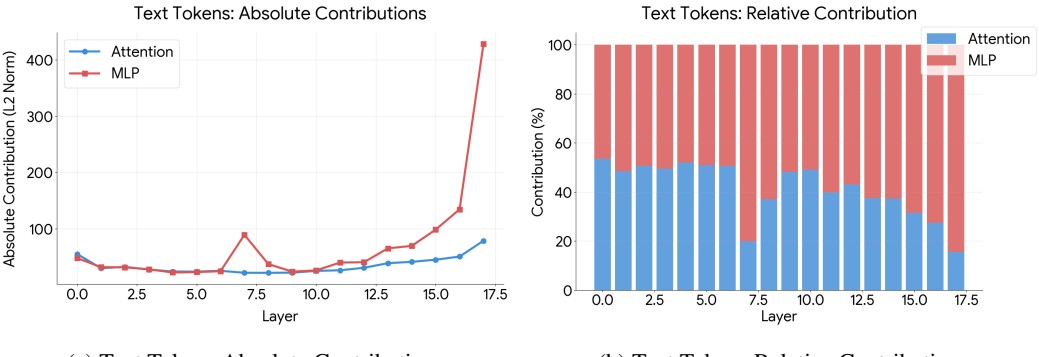

(a) Text Tokens Absolute Contributions

(b) Text Tokens Relative Contributions

Figure 6: Attention and MLP contributions to text token updates: Relative contributions reveal attention and MLP contribute roughly equally throughout most layers, with attention comprising 40-60% of total changes. Absolute contributions shows that MLP spikes at Layer 7 to an L2 Norm of 100 before stabilizing, then surges once more at the final layer to 400.

## B.2 QWEN2

Next, we replicate these experiments in QWEN2-VL-2B-INSTRUCT. In both Figure 7 and Figure 9, attention is the major contributor to the residual stream. This aligns with PALIGEMMA's distribution whose main contributions are sourced from attention. However, attention in PALIGEMMA only overtakes MLP contribution by a slight amount, while the distribution for QWEN2 shows a much larger degree of favorability towards attention. In accordance to vision tokens, MLP contribution in QWEN2 has a small spike in the very early layers and a large spike in late layers. This shows when MLP processes visual features: in very early layers, the raw data is marked. During the late layers, MLP finally transforms the data during cross-modal integration. This differs from Figure 6 in that PALIGEMMA spikes in MLP in early-to-intermediate layers, rather than in the second layer such as QWEN2.

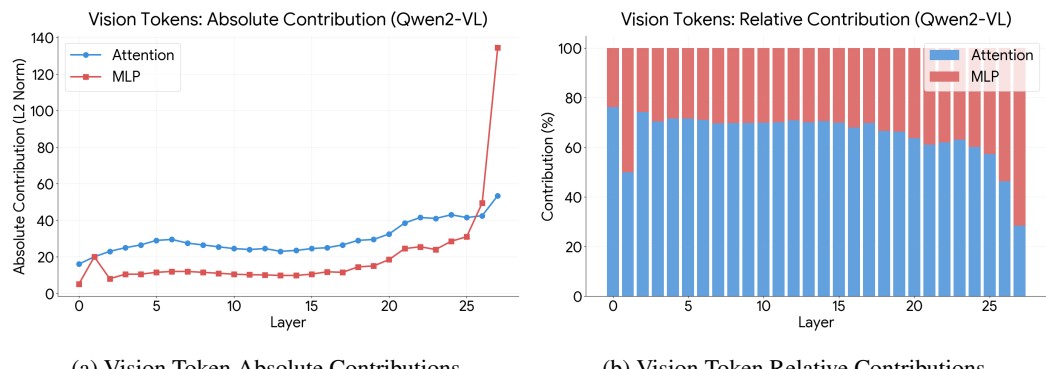

(a) Vision Token Absolute Contributions

(b) Vision Token Relative Contributions

Figure 7: QWEN2's Attention and MLP contributions to vision token updates: Relative contributions reveal attention consistently contributes more than MLP to the residual stream in early to intermediate layers, with a small spike in MLP during the very early layers. In the late layers, the MLP contributes significantly to the residual stream.

In Figure 9, MLP's contribution to the residual stream spikes in late layers. These results are similar to those found in PALIGEMMA's contributions: textual features are integrated alongside visual features for the final output. However, in contrast to PALIGEMMA's results, attention contributes to a higher degree overall, reaching nearly 60%-70% in the late layers.

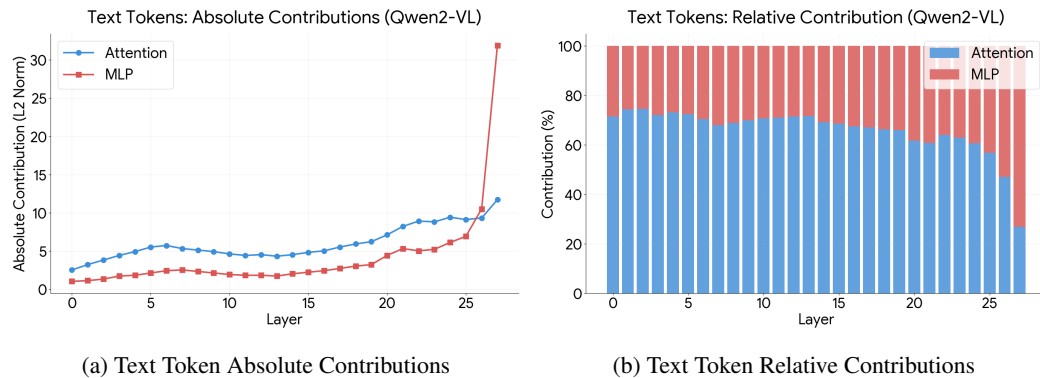

(a) Text Token Absolute Contributions
(b) Text Token Relative Contributions

Figure 8: QWEN2's Attention and MLP contributions to text token updates: Relative contributions reveal attention consistently contributes more than MLP to the residual stream in early to intermediate layers. In late layers, MLP drastically contributes to the residual stream.

## B.3    PALIGEMMA2-10B

Finally, we determine relative and absolute contributions on a PALIGEMMA model scaled to 10B parameters. Despite using the same vision encoder, the attention and MLP contributions shown in the vision token results are abnormal compared to the results seen in PALIGEMMA-3B and QWEN2. Both MLP and attention contributions contribute at high rates in early layers, with MLP being the priority contributor. However, in subsequent layers, attention becomes the majority contributor. This is in contrast to the 2B and 3B parameter models which start off low in contributions to the residual stream, with a small rise in attention and a large spike in MLP at the very last layer.

The text tokens result in similarly bizarre behavior in that contributions start off high and diminish overtime. However, attention spikes at the intermediate layers. Distribution-wise, attention takes over 90% of the contributions to the residual stream consistently. This is unlike the 2B and 3B models which spiked in MLP contributions at late layers and PALIGEMMA2-10B's distribution which spiked in MLP contributions at the early layers.

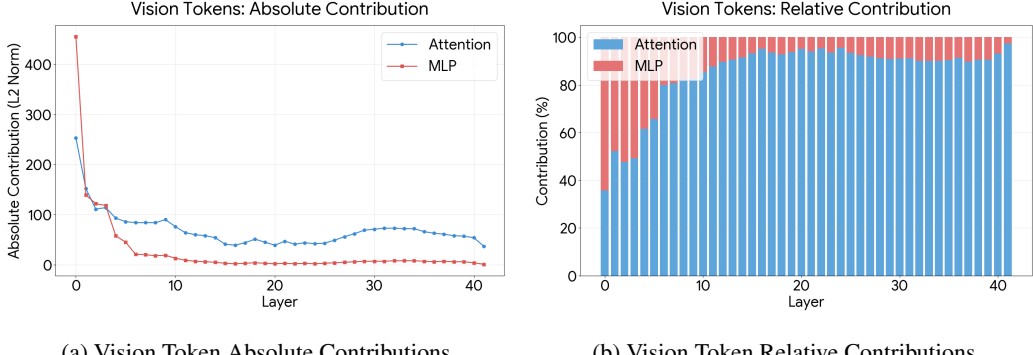

(a) Vision Token Absolute Contributions
(b) Vision Token Relative Contributions

Figure 9: PALIGEMMA2-10B's Attention and MLP contributions to vision token updates: Relative contributions reveal attention consistently contributes more than MLP to the residual stream in intermediate to late layers. In early layers, MLP drastically contributes to the residual stream.

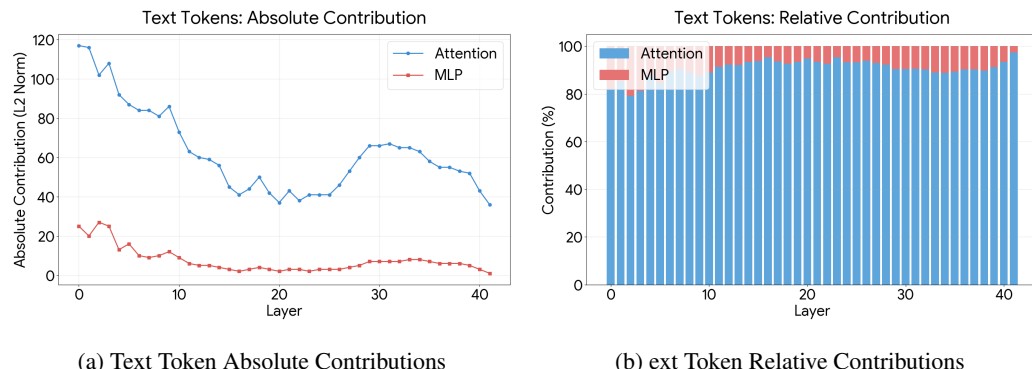

(a) Text Token Absolute Contributions    (b) ext Token Relative Contributions

Figure 10: PALIGEMMA2-10B's Attention and MLP contributions to text token updates: Relative contributions reveal attention contributes more than MLP to the residual stream consistently. While contributions to the residual stream is shown to diminish, attention spikes in layers 25-30.

## C   ATTENTION CONTRIBUTIONS VISUALIZED

### C.1   PALIGEMMA

For the purposes of this experiment, we compare the raw scores from layer 10, an intermediate layer, to get a better understanding of the processing distribution. Since they are a decomposition of the attention contributions, they are small compared to the total composition. Figure 11 shows that image token queries primarily attend to image token keys rather than incorporating information from text queries. Similarly, text token queries primarily attend to text token keys, with low attention from image queries. This aligns with the "process-then-retrieve" workflow, which defers information integration across modalities in intermediate layers.

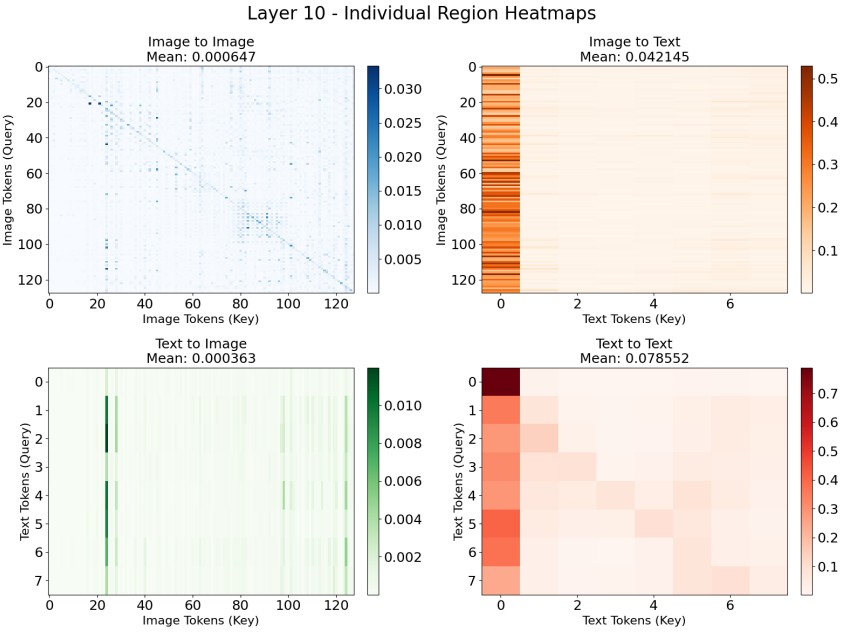

Figure 11: Attention score decomposition for PALIGEMMA-3B: Each quadrant shows attention from query tokens (rows) to key tokens (columns). Top-left: Vision queries attending to vision keys (V to V, mean=0.000647). Top-right: Vision queries attending to text keys (V to T, mean=0.000363). Bottom-left: Text queries attending to vision keys (T to V, mean=0.042145). Bottom-right: Text queries attending to text keys (T to T, mean=0.078552).

## C.2 QWEN2

This experiment was repeated with the QWEN2-VL-2B-INSTRUCT model. Similar to the experiment performed in PALIGEMMA, Figure 13 demonstrates that each query token primarily attends to a key of the same modality. As further evidence, the image token queries do not attend to the text token keys at all, indicating the lack of cross-modal integration towards text token keys. This signals a structural bias, in which both vision and text queries attend to image keys while only text queries attend to text keys. Broadly, this suggests that QWEN2 has a textual bias, allowing cross-modal integration to occur in only one direction. On a lesser note, the image queries attend to the image keys more than the text queries do. However, the mean scores still lie within a similar range of each other. The relatively equal distribution, in which both modalities are emphasized when attending to image token keys, further signals model reliance on text information for visual output.

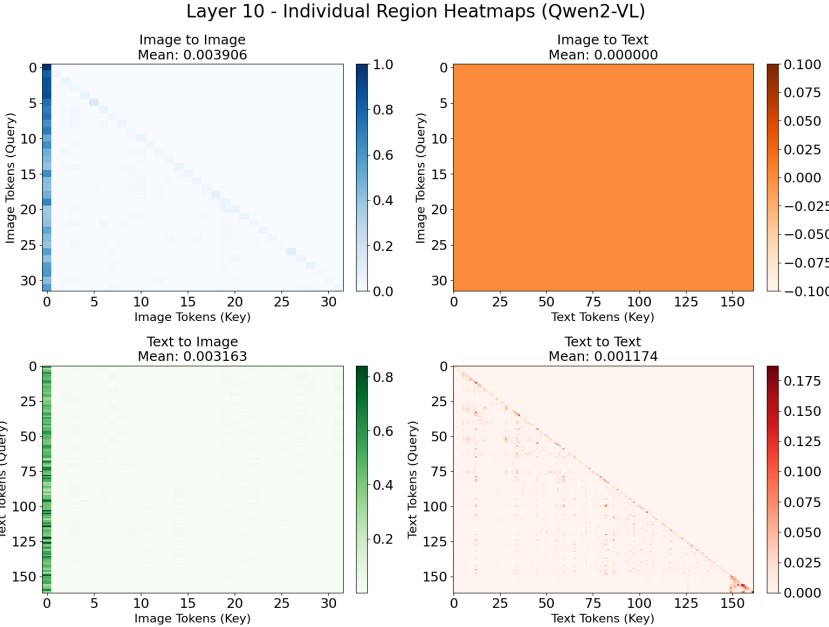

Figure 12: Attention score decomposition for QWEN2-VL-2B: Each quadrant shows attention from query tokens (rows) to key tokens (columns). Top-left: Vision queries attending to vision keys (V to V, mean=0.003906). Top-right: Vision queries attending to text keys (V to T, mean=0). Bottom-left: Text queries attending to vision keys (T to V, mean=0.003163). Bottom-right: Text queries attending to text keys (T to T, mean=0.001174).

## C.3 PALIGEMMA2-10B

Next, we test how query attention distributes to keys of different modalities in a scaled model. Since it is part of the PALIGEMMA family, we expect to see the results will follow an attention trend similar to Figure 11. As expected, image token keys are derive most of its attention from image queries rather than the text queries, and vice versa for text token keys. However, it is notable that text keys are attended to at large rates by both image and text queries. Image queries have a mean attention score that is greater than half, while the score as attended to by text queries is greater than 0.90.

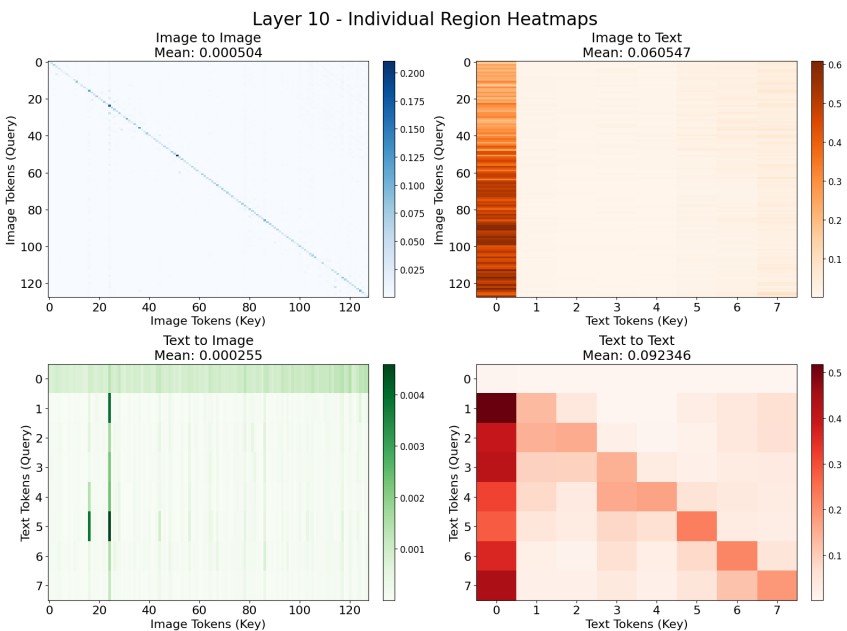

Figure 13: Attention score decomposition for PALIGEMMA2-10B: Each quadrant shows attention from query tokens (rows) to key tokens (columns). Top-left: Vision queries attending to vision keys (V to V, mean=0.000504). Top-right: Vision queries attending to text keys (V to T, mean=0.060547). Bottom-left: Text queries attending to vision keys (T to V, mean=0.000255). Bottom-right: Text queries attending to text keys (T to T, mean=0.092346).

# D  PATCHING EVALUATION METRICS

To quantify the causal effect of activation patching on model performance, we track two primary metrics focused on the ground-truth target token $y^*$ (e.g., the correct count or object name). Let $\mathcal{L}_{\text{CE}}$ denote the cross-entropy loss and $z_{y^*}$ denote the pre-softmax logit corresponding to the target token.

## D.1  CLEAN VS. PATCHED LOSS

The Clean Loss serves as the baseline performance of the model on the unaltered input $x$. The Patched Loss measures the performance after an intervention is applied, where activations from a source layer are injected into a target layer.

$$\text{Loss}_{\text{clean}} = -\log P(y^* \mid x) \tag{1}$$

$$\text{Loss}_{\text{patched}} = -\log P(y^* \mid x_{\text{patched}}) \tag{2}$$

A distinct increase in Patched Loss compared to Clean Loss indicates that the intervened representation contained critical information required for the correct prediction.

## D.2  LOGIT DIFFERENCE ($\Delta$ LOGIT)

While loss measures probability, the raw logit provides a more linear measure of the model's confidence before the softmax saturation. We report the change in the target token's logit:

$$\Delta\text{Logit} = z_{y^*}^{(\text{patched})} - z_{y^*}^{(\text{clean})} \tag{3}$$

A negative $\Delta$ Logit indicates that the patch disrupted the mechanism responsible for predicting the correct answer, while a positive $\Delta$ Logit suggests the patch reinforced the correct prediction.

# E   ATTENTION-OUTPUT PATCHING EXPERIMENTS ON QWEN2-VL

We perform attention-only activation patching in QWEN2-VL to determine the role of the attention mechanism in the model. We isolate the self-attention submodule in the Vision and Text layers. With the same patching configuration as illustrated in PALIGEMMA, we extract the hidden states output from the attention mechanism of the "source layers" and inject them into the attention output of "target layers." The MLPs of the target layers were left untouched, and we continued to process the residual stream naturally. Patching all activations from early to late layers, rather than only the attention activation outputs, shows the model's reliance on the MLPs.

## E.1   TEXTUAL RESULTS

Front-layer activation patching in QWEN2, as seen in Appendix G.2.2, demonstrates that skipping text layers results in a massive logit drop, compared to the much smaller drop observed in most configurations when patching only attention. This confirms that the MLPs help adapter-based VLMs process and reason the information provided by textual captions. The -1.41 drop from early to late layers shows that attention is not redundant. Instead, the model determines what tokens to attend to, from vision to answer tokens, as it progresses throughout the layers. The MLPs actually perform the computation as prompted, which in this case was counting. Attention patching remained mainly unaffected.

## E.2   VISION RESULTS

The vision encoder typically displayed high redundancy in its attention patterns. There is a near-zero impact of patching early attention into mid layers, and the minor drop when patching early attention into late layers suggests that the models' attention maps are established in the very early layers. The subsequent attention layers mainly reinforce the notion of where the model "should look" without adding much additional information. Once the model identifies the relevant object regions in the early layers, the attention module's job is complete. The deeper layers most likely use their MLPs to extract features from those attended regions.

Table 2: Layer-wise Attention Patching Results for Qwen2-VL

| Modality | Patch Direction | Clean | Patched | $\Delta$ Logit | Clean Loss | Patched Loss |
|----------|-----------------|-------|---------|----------------|------------|--------------|
| Vision | Early $\rightarrow$ Mid | 15.79 | 15.82 | +0.04 | 19.57 | 19.92 |
| Vision | Late $\rightarrow$ Mid | 15.79 | 15.52 | -0.27 | 19.57 | 20.68 |
| Vision | Early $\rightarrow$ Late | 15.77 | 15.10 | -0.67 | 19.58 | 19.87 |
| Vision | Mid $\rightarrow$ Late | 15.77 | 15.22 | -0.55 | 19.58 | 19.78 |
| Text | Early $\rightarrow$ Mid | 15.77 | 14.73 | -1.04 | 19.58 | 20.14 |
| Text | Late $\rightarrow$ Mid | 15.79 | 16.02 | +0.23 | 19.57 | 20.12 |
| Text | Early $\rightarrow$ Late | 15.79 | 14.37 | -1.41 | 19.57 | 17.18 |
| Text | Mid $\rightarrow$ Late | 15.79 | 14.89 | -0.90 | 19.58 | 14.95 |

# F   ATTENTION-OUTPUT PATCHING EXPERIMENTS ON PALIGEMMA-10B

## F.1   TEXTUAL RESULTS

Similarly, we isolate the output of the Text Attention heads in PALIGEMMA-10B. Patching text attention yields generally high results with a massive difference between the average clean and patched losses. Specifically, the mid to late attention patching resulted in the largest $\Delta$logit difference of +5.79, and following it is the $\Delta$logit difference from the early attention layers patched to the late layers with a $\Delta$logit difference of +5.12. This suggests that the information extracted by the attention heads in the early and middle layers are the most critical component of the final answer generation. However, information propagated from late layers into the middle layers results in a slight $\Delta$ logit decrease, indicating that information derived in the final layers cannot be successfully integrated back into the middle layers.

## F.2 VISION RESULTS

When patching attention outputs from early (+2.06) and mid (+2.12) layers to late layers, there is a slight, similar increase in the $\Delta$logit. Their nearly identical difference suggests that visual attention doesn't evolve dramatically throughout the network. While the model still requires visual context to function, the magnitude of improvement is significantly lower than that of textual attention patching implies that the final classification decision occurs within the text attention heads. When patching later attention outputs into the mid layers, there is a negligible change, unlike in the text stream where back patching was more harmful to the model's performance.

Table 3: Layer-wise Attention Patching Results for PaliGemma2-10B

| Modality | Patch Direction | Clean | Patched | $\Delta$ Logit | Clean Loss | Patched Loss |
|---|---|---|---|---|---|---|
| Vision | Early $\to$ Mid | 12.84 | 13.34 | +0.50 | 12.26 | 11.74 |
| Vision | Late $\to$ Mid | 12.84 | 12.86 | +0.02 | 12.26 | 12.13 |
| Vision | Early $\to$ Late | 12.85 | 14.90 | +2.06 | 12.26 | 9.05 |
| Vision | Mid $\to$ Late | 12.85 | 14.97 | +2.12 | 12.26 | 9.72 |
| Text | Early $\to$ Mid | 12.85 | 18.36 | +5.51 | 12.26 | 6.77 |
| Text | Late $\to$ Mid | 12.85 | 12.24 | -0.61 | 12.26 | 12.80 |
| Text | Early $\to$ Late | 12.84 | 17.96 | +5.12 | 12.26 | 11.17 |
| Text | Mid $\to$ Late | 12.84 | 18.63 | +5.79 | 12.26 | 6.22 |

## G ACTIVATION FRONT PATCHING EXPERIMENTS

### G.1 PALIGEMMA

#### G.1.1 EXPERIMENTAL SETUP

We perform front-activation patching experiments to compare textual and visual representations across various layer groups in PALIGEMMA. For each image, we run two forward passes: a clean and a patched run. During the clean run, the model processes the image-text input normally, while hidden activations from selected layers are recorded. During the patched run, or the second forward pass, the hidden activations in the target layers are replaced with the activations saved from the earlier source layers in the clean run. For textual front patching, we intervene on the transformer layers of PALIGEMMA. Activations from early or mid textual layers, the source layers, are saved during the clean pass and injected into the last three textual layers, target layers, before the output layer, during the patched pass. For vision front patching, we apply the same methodology to the encoder layers of the vision tower, injecting activations from earlier vision layers into later ones. Across both modalities, we isolate the textual and vision representations to properly analyze their results. Performance is measured using the logit assigned to the ground-truth answer token at the final position and the corresponding cross-entropy loss. The various source layer groupings we tested front patching on were early layers of 2, 3, 4, mid layers of 7, 8, 9, and late layers of 10, 11, 12,

#### G.1.2 TEXTUAL REPRESENTATION PATCHING RESULTS

In textual front patching, the performance typically decreases across various groups of source patching. Early-to-late patching reduces target logits by more than 10 points on average, while cross-entropy loss increases sharply to over 30. In general, any sort of front patching, early, mid, and late to the final layers, all result in a sharp increase in loss. This indicates that textual representations are not interchangeable across layers and that later layers encode essential information for reasoning and answer generation.

#### G.1.3 VISION REPRESENTATION PATCHING RESULTS

On the contrary, front patching within the visual representations produces weaker results. Early vision patching reduces the target logit but increases the loss slightly, indicating that PALIGEMMA can still function. When patching mid- and late-vision layers, the target logit and cross-entropy loss continue to decrease relative to the clean run results. Later vision-layer activations aren't critical to

the model's reasoning and answer-generation phase. Intervening at these layers only yields lower-entropy predictions. We suggest that visual representations are partially redundant and become more abstract as the depth increases.

### G.1.4  MODALITY ASYMMETRY

Overall, textual representations are depth-sensitive and essential in PALIGEMMA, while visual representations are depth-dependent and have a weaker influence. Visual information isn't an independent factor in the model's final prediction; rather, it's a signal of the biases and logic in the textual stream. Later vision-layer activations cannot serve alone and require multi-modal alignment.

Table 4: Layer-wise Representation Patching Results for PaliGemma-3B

| Modality | Layers | Clean | Patched | $\Delta$ Logit | Clean Loss | Patched Loss |
|---|---|---|---|---|---|---|
| Vision | 2–4 | 12.79 | 11.79 | -0.99 | 14.86 | 14.70 |
| Vision | 5–7 | 12.79 | 12.21 | -0.58 | 14.86 | 13.98 |
| Vision | 7–9 | 12.79 | 12.50 | -0.28 | 14.86 | 13.36 |
| Vision | 10–12 | 12.79 | 12.40 | -0.39 | 14.86 | 13.25 |
| Text | 2–4 | 12.76 | -0.75 | -13.51 | 14.87 | 23.64 |
| Text | 5–7 | 12.79 | 2.62 | -10.17 | 14.86 | 32.25 |
| Text | 7–9 | 12.76 | 0.28 | -12.48 | 14.87 | 33.49 |
| Text | 10–12 | 12.76 | 0.92 | -11.84 | 14.87 | 25.98 |

### G.2  QWEN2-VL

### G.2.1  EXPERIMENTAL SETUP

Within the QWEN2-VL-2B-INSTRUCT architecture, we perform Front Patching, similar to PALIGEMMA, by extracting internal activation states from earlier layers and injecting them into the final three layers of the model.

### G.2.2  TEXTUAL REPRESENTATION PATCHING RESULTS

By patching different configurations of layers (2-4, 5-7, 7-9, and 10-12) into the last three layers, the language model of QWEN2-VL severely collapsed. Across all configurations, the logit score for the correct answer dropped from a highly confident score of 15.79 to around 1.0-2.3. The LLM's residual stream experiences don't benefit from patching and require linear transformation. The extreme logit drop reveals that the "concept" of the target logit "2" is not retrieved immediately. The model likely spends the intermediate layers integrating the visual tokens with the text prompt, which would've increased the probability of the counting token. Patching across these configurations underscores the importance of the reasoning phase, where both modalities align.

### G.2.3  VISION REPRESENTATION PATCHING RESULTS

Patching earlier layers of the vision tower into the final layer improved the logit score by approximately 0.65, which is a slight performance increase. However, for other source layer configurations, there were negligible performance drops, maintaining the model's confidence in the correct answer. This suggests that the fundamental features required for "counting" (what the model was specifically prompted with) are established in the very early layers of the encoder.

Table 5: Layer-wise Representation Patching Results for Qwen2-VL

| Modality | Layers | Clean | Patched | $\Delta$ Logit | Clean Loss | Patched Loss |
|----------|--------|-------|---------|----------|------------|--------------|
| Vision | 2–4 | 15.78 | 16.42 | +0.65 | 19.58 | 20.18 |
| Vision | 5–7 | 15.78 | 15.49 | -0.29 | 19.58 | 20.33 |
| Vision | 7–9 | 15.79 | 15.40 | -0.39 | 19.57 | 20.20 |
| Vision | 10–12 | 15.77 | 15.49 | -0.28 | 19.58 | 20.60 |
| Text | 2–4 | 15.79 | 2.03 | -13.76 | 19.57 | 12.09 |
| Text | 5–7 | 15.79 | 1.82 | -13.97 | 19.57 | 10.09 |
| Text | 7–9 | 15.79 | 1.01 | -14.77 | 19.57 | 11.24 |
| Text | 10–12 | 15.79 | 2.29 | -13.50 | 19.57 | 12.00 |

## G.3 PALIGEMMA-10B

We conduct residual stream activation patching on 82 valid image samples in PALIGEMMA2-10B-PT-448 to measure the semantic stability of visual and textual tokens throughout the model's 42 layers. We perform a front patching experiment from layers 2-6 and layers 18-22 onto the final layers, 38-41.

### G.3.1 TEXTUAL REPRESENTATION PATCHING RESULTS

The text stream was revealed to be extremely sensitive, as shown by the drop in the $\Delta$ logit from -16.70 when patching from early to late layers. The textual representations in earlier layers are incompatible with those in later layers, indicating that the model requires the full length of the network to fully "process" the text tokens needed for answer generation.

### G.3.2 VISION REPRESENTATION PATCHING RESULTS

In stark comparison, the vision tokens remain linearly stable throughout the network. Patching visual states from early or mid layers into later layers resulted in a similar performance improvement. This indicates that early and mid-layer visual representations are already semantically usable for the final classification layer. Especially for earlier layers, this implies that skipping intermediate processing layers may provide the output heads with a better visual signal.

Table 6: Layer-wise Representation Patching Results for PaliGemma2-10B

| Modality | Layers | Clean | Patched | $\Delta$ Logit | Clean Loss | Patched Loss |
|----------|--------|-------|---------|----------|------------|--------------|
| Vision | 2–6 | 12.84 | 16.35 | +3.51 | 12.26 | 9.77 |
| Vision | 18–22 | 12.85 | 16.44 | +3.59 | 12.26 | 10.03 |
| Text | 2–6 | 12.84 | -3.86 | -16.70 | 12.26 | 76.71 |
| Text | 18–22 | 12.84 | 12.63 | -0.21 | 12.26 | 11.64 |

## H FORMAL METRIC DEFINITIONS

We formally define the metrics used to evaluate the models' cross-modal behavior. Let $L$ be the number of transformer layers; $H^{(l)} \in \mathbb{R}^{T \times d}$ denote the hidden state activations at layer $l$, where $T$ is the sequence length and $d$ is the hidden dimension; and $I_{\text{vis}}$ and $I_{\text{text}}$ denote the sets of indices corresponding to vision and text tokens, respectively.

### H.1 COSINE SIMILARITY OF REPRESENTATIONS

To measure the stability of visual representations across consecutive layers, we compute the cosine similarity between the hidden state of a token at layer $l$ and layer $l + 1$. For a specific vision token at position $t \in I_{\text{vis}}$, the similarity is defined as:

$$\text{CosSim}(t, l) = \frac{H_t^{(l)} \cdot H_t^{(l+1)}}{\|H_t^{(l)}\|_2 \|H_t^{(l+1)}\|_2} \quad (4)$$

The reported metric is the average similarity over all vision tokens across $N$ samples in the Pixel-Prose dataset:

$$\text{AvgSim}^{(l)} = \frac{1}{N} \sum_{k=1}^{N} \left( \frac{1}{|I_{\text{vis}}|} \sum_{t \in I_{\text{vis}}} \text{Sim}_k(t, l) \right) \tag{5}$$

## H.2 RESIDUAL STREAM UPDATE & DECOMPOSITION

The transformer residual stream at layer $l$ is updated by two distinct sub-modules: Multi-Head Attention (Attn) and the Multi-Layer Perceptron (MLP). Within the transformer stack of an adapter-based VLM (assuming a standard pre-LN configuration), the hidden state update is defined as:

$$H^{(l)} = H^{(l-1)} + \Delta_{\text{Attn}}^{(l)} + \Delta_{\text{MLP}}^{(l)} \tag{6}$$

where the individual contributions are:

$$\Delta_{\text{Attn}}^{(l)} = \text{Attn}(\text{LN}(H^{(l-1)})) \tag{7}$$

$$\Delta_{\text{MLP}}^{(l)} = \text{MLP}(\text{LN}(H^{(l-1)} + \Delta_{\text{Attn}}^{(l)})) \tag{8}$$

The **Residual Stream Update** metric quantifies the magnitude of change introduced by layer $l$. We calculate the $L_2$ norm of the total difference between the pre-layer and post-layer representations for a specific modality set $S \in \{I_{\text{vis}}, I_{\text{text}}\}$:

$$\text{Update}_S^{(l)} = \frac{1}{|S|} \sum_{t \in S} \|H_t^{(l)} - H_t^{(l-1)}\|_2 \tag{9}$$

To determine the drivers of this processing, we calculate the **Relative Contribution** of each sub-module:

$$\text{Contrib}_{\text{Attn}}^{(l)} = \frac{\|\Delta_{\text{Attn}}^{(l)}\|_2}{\|\Delta_{\text{Attn}}^{(l)}\|_2 + \|\Delta_{\text{MLP}}^{(l)}\|_2} \tag{10}$$

## H.3 ATTENTION BY MODALITY

We analyze the attention distribution of the final token (the query $q_{last}$), which is responsible for generating the next token. Let $A_{last,j}^{(l)}$ represent the attention weight from the last token to key token $j$ at layer $l$, averaged across all attention heads $h$:

$$A_{last,j}^{(l)} = \frac{1}{N_{heads}} \sum_{h=1}^{N_{heads}} \text{Softmax} \left( \frac{Q_{last}^{(l,h)} (K_j^{(l,h)})^T}{\sqrt{d_k}} \right) \tag{11}$$

We define the **Modality Attention Score** as the total probability mass allocated to a specific modality (Vision or Text):

$$\text{Attn}_{\text{vision}}^{(l)} = \sum_{j \in I_{\text{vis}}} A_{last,j}^{(l)}, \qquad \text{Attn}_{\text{text}}^{(l)} = \sum_{j \in I_{\text{text}}} A_{last,j}^{(l)} \tag{12}$$

## H.4 AVERAGING PROCEDURES

All reported results are averaged over the PixelProse dataset. For any metric $M$ computed for a single sample $k$, the final reported value is:

$$\bar{M} = \mathbb{E}_{x \sim \mathcal{D}}[M(x)] \approx \frac{1}{N} \sum_{k=1}^{N} M_k \tag{13}$$

Where $N = 100$ for our representational analysis experiments.

