# OpenReview forum: "Process-then-Retrieve: A Mechanistic Study of Cross-Modal Alignment in Vision-Language Models"
_ICLR.cc/2026/Workshop/Sci4DL — Sci4DL 2026_

### Official Review · Reviewer_BwGU · 2026-02-24

**Fit:** 3
**Significance:** 2
**Confidence:** 2

**Summary:**

This paper investigates how VLMs integrate textual and visual information. Their main claim is that most of the visual information is injected in the later layers with the earlier layers simply processing the textual inputs. The authors support this with cosine similarity of visual embeddings across different layers, attention-weight analyses,  and several activation-patching interventions, concluding that cross-modal retrieval happens only in the final few blocks, claiming that this is what leads to the observed textual dominance in modern day VLMs.

**Strengths:**

1. The paper proposes a hypothesis of "process-then-retrieve" and then provides several empirical diagnostic results such as successive visual embeddings cosine similairity, attention-output patching, and residual stream update (how the magnitude of visual embeddings changes pre and post norm across the stack)
2. The paper includes results across two multi-modal model families PaliGemma and Qwen-VL strenthenging the claims validity.
3. The paper clearly describes all metrics and mathematical formulations in Appendix F which makes it easy to follow.

**Suggestions:**

1. The paper uses only a single dataset PixelProse with only a 100 samples from it. It is hard to generalize the conclusions drawn regarding different components based on just one dataset. Moreover, other than just a counting VQA task, the authors should consider increasing the datasets used in the experimental setup to covert a wide range of tasks such as Spatial and Compositional Reasoning, OCR based VQA, and Fine-Grained Attribute Recognition, etc. It will also be great if the authors can show results on general VQA datasets like MMMU or VQAv2.
2. The paper tracks mean cosine similarity , layer-wise attention distributions , and residual stream updates by averaging across their 100 sample subset of the PixelProse dataset. However, they do not condition or whether the models response was actually correct or incorrect during on these samples. This can lead to a confounding factor: its possible that the cosine similarity and attention distributions behave a certain way for correct vs incorrect samples (where the model may hallucinate etc). It is important to correct for label based contioning in experiments.
3. The analysis only holds for Adapter based models like Qwen/Pali-Gemma. But there are several other multi-modal architectures which use a more complex architecture like Resampler/Q-Former in BLIP family. It is important to conduct a similar investigation into those architectures as well.
4. Section 2.4 is under-investigated, the authors first try to support their claim by training a linear classifier over the text and image features from transformer layers and hypothesize that the text features should have a higher accuracy from earlier transformer layers and vice versa, but their experiments do not support this. The authors simply claim: "However, we observe that probe performance is highest when classifiers are evaluated on the same layers on which they were trained, and accuracy degraded when applied to earlier layers. This might be because probes assume linearity and may miss non-linear changes.". I think this should be investigated in more detail and the authors should provide some more analysis as to why this happens.

---

### Official Review · Reviewer_d3hV · 2026-02-27

**Fit:** 2
**Significance:** 2
**Confidence:** 3

**Summary:**

This paper presents a mechanistic study of adapter-based vision-language models (VLMs), specifically PaliGemma-3B and Qwen2-VL, to understand how they integrate visual and textual information. The authors propose and validate a "process-then-retrieve" workflow, where early layers focus on textual context while visual information remains largely static and segregated. Through methods like activation patching and residual stream analysis, the study reveals that significant cross-modal alignment and visual retrieval only occur in the final transformer blocks, contributing to a "textual dominance" that can limit visual reasoning.

**Strengths:**

1. The paper moves beyond simple performance observations to identify specific internal behaviors, such as the stability of visual embeddings in early layers versus the heavy early processing of text
2. The authors utilize a robust suite of interpretability tools—including Representational Similarity Analysis (RSA), attention patching, and residual stream attribution—to provide a multi-faceted view of the model's internal dynamics.

**Suggestions:**

1. The experimental evaluation focuses heavily on a VQA retrieval task (specifically object counting). It remains unclear if the "process-then-retrieve" workflow holds as strictly for more complex tasks like spatial reasoning or long-form image captioning.
2. The authors admit that their linear probing results—which showed a sudden performance spike in late layers—might be limited because probes assume linearity and may fail to capture non-linear visual transformations occurring in intermediate layers.
3. While the study is thorough for adapter-based VLMs, it does not compare these results against other architectures (such as interleaved or fused-attention models), leaving it uncertain if this is a universal VLM trait or strictly a byproduct of the linear projection adapter design.

---

### Official Review · Reviewer_g3rb · 2026-02-28

**Fit:** 3
**Significance:** 2
**Confidence:** 2

**Summary:**

This paper uses mechanistic interpretability to show that adapter-based VLMs (PaliGemma, Qwen2-VL) employ a "process-then-retrieve" workflow. The authors demonstrate that early layers focus entirely on processing text, leaving visual embeddings largely unmodified until late-stage cross-modal integration occurs in the final transformer layers.

**Strengths:**

- The paper utilizes a decent approach to mechanistic interpretability. Through considering results from cosine similarity, attention versus MLP attribution, and activation patching, the authors provide decent evidence for their claims.
- The experimental design aligns directly with the workshop's focus on empirical studies of learned representations. It isolates specific variables to identify the structural mechanisms driving the known "modality gap" in VLMs.
- Testing the hypothesis on two models from distinct model families (PaliGemma-3B and Qwen2-VL) provides evidence that the observed "process-then-retrieve" dynamic generalizes across different well-known VLMs.

**Suggestions:**

- The evaluation relies on a single VQA task. It remains unclear whether the observed late-stage visual retrieval is an inherent architectural constraint or an outcome of this specific prompt. Adding tasks that demand e.g. complex visual reasoning would clarify this distinction.
-In Section 2.4, the performance drop of late-layer linear probes when applied to early layers is attributed to the probes missing non-linear changes. This explanation appears to be rather speculative. Adding a non-linear probing baseline would clarify whether visual features are actually unrecoverable in early layers or simply encoded non-linearly.
- The description of activation patching in Section 2.2 lacks detail in the main text. Specifying the exact token positions being patched (e.g., all visual tokens vs. the final answer token) is necessary for clarity and reproducibility.

---

### Meta-Review · Area_Chair_abXW · 2026-02-28

**Recommendation:** Accept

**Metareview:**

This paper studies how vision-language models process and merge information from the two modalities across layers. The claims are clearly demonstrated through several experiments. I recommend acceptance.

---

### Decision · Program_Chairs · 2026-03-02

Accept